# Disparate Effects of Two Clerodane Diterpenes of Giant Goldenrod (*Solidago gigantea* Ait.) on *Bacillus spizizenii*

**DOI:** 10.3390/ijms25031531

**Published:** 2024-01-26

**Authors:** Zoltán Bozsó, Virág Lapat, Péter G. Ott, Ágnes M. Móricz

**Affiliations:** Plant Protection Institute, HUN-REN Centre for Agricultural Research, Herman Ottó Str. 15, H-1022 Budapest, Hungary; bozso.zoltan@atk.hun-ren.hu (Z.B.); ott.peter@atk.hun-ren.hu (P.G.O.)

**Keywords:** *Solidago gigantea* Ait. (giant goldenrod), clerodane diterpenes, *Bacillus*, *Rhodococcus*, antibacterials, transcriptome

## Abstract

New substances with antimicrobial properties are needed to successfully treat emerging human, animal, or plant pathogens. Seven clerodane diterpenes, previously isolated from giant goldenrod (*Solidago gigantea*) root, were tested against Gram-positive *Bacillus subtilis*, *Bacillus spizizenii* and *Rhodococcus fascians* by measuring minimal bactericidal concentration (MBC), minimal inhibitory concentration (MIC) and half-maximal inhibitory concentration (IC_50_). Two of them, Sg3a (a dialdehyde) and Sg6 (solidagoic acid B), were proved to be the most effective and were selected for further study. *Bacillus spizizenii* was incubated with the two diterpenes for shorter (1 h) or longer (5 h) periods and then subjected to genome-wide transcriptional analyses. Only a limited number of common genes (28 genes) were differentially regulated after each treatment, and these were mainly related to the restoration of cell membrane integrity and to membrane-related transports. Changes in gene activity indicated that, among other things, K^+^ and Na^+^ homeostasis, pH and membrane electron transport processes may have been affected. Activated export systems can be involved in the removal of harmful molecules from the bacterial cells. Inhibition of bacterial chemotaxis and flagellar assembly, as well as activation of genes for the biosynthesis of secondary metabolites, were observed as a general response. Depending on the diterpenes and the duration of the treatments, down-regulation of the protein synthesis-related, oxidative phosphorylation, signal transduction and transcription factor genes was found. In other cases, up-regulation of the genes of oxidation–reduction processes, sporulation and cell wall modification could be detected. Comparison of the effect of diterpenes with the changes induced by different environmental and nutritional conditions revealed several overlapping processes with stress responses. For example, the Sg6 treatment seems to have caused a starvation-like condition. In summary, there were both common and diterpene-specific changes in the transcriptome, and these changes were also dependent on the length of treatments. The results also indicated that Sg6 exerted its effect more slowly than Sg3a, but ultimately its effect was greater.

## 1. Introduction

There is a constant demand for new antimicrobial compounds in the fields of human and animal health, as well as plant protection. This is necessary, among other things, for effective control of newly emerging antibiotic-resistant microorganisms and for the development of environmentally friendly control methods against them [1]. Among the various possible sources of new antibiotics, plant-derived molecules could be one solution. Since plants are constantly exposed to the attack of various pathogens, they produce a wide variety of secondary metabolites, which are rich sources of active ingredients that plants can use in defense reactions [2,3].

*Solidago gigantea* Ait. (giant goldenrod) originated from North America and is an invasive weed in the whole of Europe. The plant is about 0.5–2.5 m tall and has persistent rhizomes [4,5,6]. Giant goldenrod produces a variety of substances with biological effects, including terpenoids, phenolic components, coumarins, essential oils and saponins [7,8,9,10,11,12,13]. Its hydrodistilled volatile oil contains more than 90 components [14]. The root and leaf contain large amounts of clerodane-type furanoid diterpenes such as solidagoic acids [12,13]. The antibacterial, antifungal and insecticidal effects of various *S. gigantea* extracts have also been demonstrated [10,15,16,17,18].

Terpenes are a structurally diverse and large group of metabolites, and thousands of different terpenes have been characterized to date. Terpenes consist of isoprene units that can be modified by cyclisation reactions. Diterpenes have a 20-carbon backbone built from four isoprene units. They can also be found in plants, fungi, bacteria and animals. In plants, diterpenes are synthesized in plastids via the methylerythritol phosphate pathway (MEP). They are based on the geranyl-geranyl-diphosphate molecule (GGPP, C-20), which is formed by joining an isopentenyl-diphosphate (C5) and a farnesyl-diphosphate (C15). Additional compounds are formed from GGPP by diterpene synthases and cytochrome P450 enzymes [19]. Clerodane diterpenes are bicyclic diterpenoids found in several hundreds of plant species. Several different biological activities have been attributed to clerodane diterpene treatments, such as antitumor, anti-inflammatory, insect antifeedant, antibacterial, antifungal and antiplasmodial effects [20]. Different clerodane diterpenes isolated from various plants were effective against both Gram-positive and Gram-negative bacteria [21,22]. In our previous papers, eight antimicrobial clerodane diterpenes were isolated and characterized from the root and four from the leaf of *S. gigantea* [12,13].

According to studies published so far, terpenes, including diterpenes, can influence the activity of bacteria in different ways [23,24]. Terpenes have been shown to destroy cell membranes and to disrupt the integrity of the outer membrane [25,26,27]. Several terpenoids were found to inhibit bacterial oxygen uptake and oxidative phosphorylation. This effect is believed to be due to their amphipathic property, which promotes their incorporation into the membrane, thereby changing fluidity and permeability, inhibiting respiration and disrupting ion transport processes [28,29,30]. In addition, terpenes may induce oxidative stress in bacterial cells [31]. Terpenes can also inhibit or modulate the activity of efflux pumps [32]. Furthermore, some of them can reduce cell adherence ability to surfaces and suppress bacterial biofilm formation [33,34].

Previously, eight different clerodane diterpenes were isolated from giant goldenrod roots [12]. Each contains a furan ring, and high-performance thin-layer chromatography–direct bioautography assays showed that they inhibit Gram-positive *B. subtilis*. In this article, we examined the antibacterial effects of seven of the eight diterpenes in more detail. We compared the various inhibitory and bactericidal concentrations of these compounds on bacteria. Two of them (Sg3a and Sg6) were found to be more effective than others. Very limited data are available in the literature on the effects of diterpenes on the bacterial transcriptome, especially on the *Bacillus* transcriptome. We do not know what processes in bacteria are affected by the different diterpenes and how they change during treatments. As genome-wide transcriptomic experiments may provide important and detailed data on bacterial responses, we performed RNA-seq experiments with *Bacillus spizizenii*. We searched for answers to the following questions during the experiments: (i) Do the two diterpenes elicit different responses in bacteria (i.e., which genes change their activity and to what extent as a result of diterpene treatments)? (ii) Which of the changes are common and which are specific? (iii) How does the reaction of bacteria change over time? (iv) What stress-related processes are induced in bacteria as a result of diterpene treatments? To this end, we compared our results with those obtained under different environmental and nutritional conditions.

## 2. Results and Discussion

### 2.1. Antibacterial Effects of Different Diterpenes Isolated from Giant Goldenrod Root

In our previous study, eight clerodane type diterpenes were isolated from giant goldenrod root: the glycol Sg1 (known as kingidiol), the epoxy-hemiacetal Sg2, the dialdehyde Sg3a, the clerodane lactone Sg3b (known as hautriwa lactone), the alcoholic substance Sg3c, the hemiacetal Sg4 and the acids Sg5 and Sg6 (known as solidagoic acid A and B, respectively) [12].

In the present study, the antibacterial effect seven of the eight previously isolated diterpenes was further tested on Gram-positive bacteria such as *B. subtilis*, *B. spizizenii* and *R. fascians* (the amount of Sg3c was not sufficient for the subsequent experiments). *B. subtilis* is a facultative anaerobe spore-forming bacterium, and, besides its industrial importance, it is the model organism of Gram-positive bacteria. *B. spizizenii* is a member of the *B. subtilis* species complex [35]. *R. fascians* is a phytopathogenic bacterium with a broad host range and causes leafy gall disease [36].

The minimal bactericidal concentration (MBC), minimal inhibitory concentration (MIC) and half-maximal inhibitory concentration (IC_50_) values of each isolate were determined. Based on the IC_50_ values, *B. subtilis* and *B. spizizenii* showed a somewhat different but tendentially similar response to the isolates. For these strains, Sg3a and Sg6 proved to be the most effective substances, and the IC_50_ values of these diterpenes were about 5.5–5.7 times higher than the IC_50_ value of the positive control gentamicin. In the case of Sg3b, the concentration used was insufficient to determine the IC_50_ inhibitory value. As against the *Bacillus* strains, Sg3a and Sg6 were the most active substances against *R. fascians*, giving an even lower IC_50_ value than gentamicin. Sg3b, Sg4 and Sg5 had also relatively low IC_50_ values (Table 1). The measured MIC and MBC values corroborated our conclusion about Sg3a and Sg6 being the best antibacterials for *Bacillus* and *R. fascians*. Similar to gentamycin, for *Bacillus* strains, the MIC and MBC values of the diterpenes (except for Sg4) were the same, which confirmed the strong bactericidal activity of these molecules (Table 2).

### 2.2. Genome Wide Transcriptome Analysis Induced by Sg3a and Sg6 Diterpenes

In addition to the characterization and structural analysis of antimicrobial substances, it is also necessary to analyze their mechanism of action. Mapping the mechanism of action is essential to be able to use them later as lead molecules in the development of an active ingredient. To study the effect of the two most effective diterpenes Sg3a and Sg6 in more detail on bacteria, genome-wide transcriptional analyses were carried out with the RNA-seq method. Sublethal concentrations of Sg3a (3.5 μg/mL) and Sg6 (1.25 μg/mL) in two different experimental settings were applied to test the effect of diterpenes on bacteria. In one case, the diterpenes were added at a lower optical density of bacterial cells (OD_600_ = 0.2) and incubated longer (Sg3a-5h, Sg6-5h and control-5h). In the other case, a higher density (OD_600_ = 0.5) bacterial population received the diterpenes for a shorter period (Sg3a-1h, Sg6-1h and control-1h). The two experimental setups represent two different aspects. While the lower bacterial number and extended incubation period show the effect of diterpenes on bacterial multiplication, a higher bacterial number and a shorter incubation period reflect the direct and rapid impact of these compounds on bacterial processes. After 5 h, the control bacteria grew from OD_600_ = 0.2 to OD_600_ = 1.34–1.51, while the Sg3a and Sg6 treated bacteria could multiply less intensively and reached OD_600_ = 0.59–0.85 and 0.81–0.92, respectively. Bacterial RNA was purified from four independent biological replicates, and, after quality and quantity checks, the RNA samples were forwarded to a prokaryotic RNA sequencing service. The filtered reads were mapped to the reference genome of *B. spizizenii* TU-B-10. The results of the mapping showed that about the same number of transcripts could be detected after all treatments. The control-5h, Sg3a-5h and Sg6-5h samples had 4064 genes in common. Within control-1h, Sg3a-1h and Sg6-1h, there were somewhat more: 4196 common genes were identified. Compared to the shared genes, only a few treatment-specific genes could be detected (Figure 1). The highest number of specific genes was found in the Sg6-5h treatment (84 genes).

The principal component analysis highlighted that the control-5h, Sg3a-5h and Sg6-5h samples separated from each other and the 1h samples, while the control-1h, Sg3a-1h and Sg6-1h samples did not separate from each other (Appendix A). The cluster analysis showed that the biological repetitions form the closest relationships, which supports the reliability of the samples. The cluster analysis also pointed out that the control-5h, Sg3a-5h and Sg6-5h samples separated from the 1h samples (Figure 2).

### 2.3. Differentially Expressed Genes in B. spizizenii after Sg3a- and Sg6-Treatments

The Sg3a- and Sg6-treated samples were compared to controls (containing ethanol as solvent of the diterpenes at equal concentrations) to detect differentially expressed genes (DEGs) after diterpene treatments (Appendix A). Without regard to the magnitude of the changes (|log_2_(FoldChange)| > 0, adjusted *p*-value < 0.05), most DEGs were found in samples incubated for longer: Sg6-5h (2019 genes) and Sg3a-5h (1583 genes), followed by those incubated shorter: Sg3a-1h (905 genes), Sg6-1h (563 genes). The number of up- and down-regulated genes did not differ considerably (Figure 3A). If we consider only genes that were activated at least 2X or repressed at least 0.5X (|log_2_(FoldChange)| > 1), we see a similar pattern with fewer genes (Sg6-5h = 1450 > Sg3a-5h = 1190 > Sg3a-1h = 493 > Sg6-1h = 315 genes) (Figure 3B). In the Sg3a-5h samples, the number of repressed genes was higher than the activated genes, indicating the domination of down-regulated processes after this treatment. On the other hand, the up-regulated genes surpassed the repressed ones numerically in the Sg6-5h samples, indicating a predominance of gene activations.

Plotting the data on volcano-type diagrams (which take into account the degree of the changes) also indicates that the most intense changes occurred in the Sg6-5h and Sg3a-5h samples, followed by Sg3a-1h, and the fewest changes were in Sg6-1h (Figure 4). These data suggest that the Sg6 diterpene exerts its effect more slowly but induces more changes in the bacterium in the longer term.

### 2.4. Overlapping B. spizizenii Genes That Changed Their Activity after Different Types of Treatments

Interestingly, only a few (28) common genes were regulated differentially after each of the four treatments (Figure 5). Of these 28 common genes, 16 were up- and two were down-regulated after all treatments (Table 3). The remaining ten shared genes show different patterns of up- or down-regulation (Appendix A). 

Overlapping genes can provide important information about the response of bacteria since these genes can reveal key processes that fundamentally influence the bacterial response to these diterpenes (Table 3). Half of the commonly activated genes are associated with the cell membrane (eight of 16). Some of them are transporters that may export harmful substances or import ions/molecules required for adaptation (four of 16). Two of these are involved in K^+^ and Na^+^ homeostasis. A high-affinity K^+^/H^+^ symporter (GYO_0650, *ydaO*/*kimA*) was induced after all types of treatments. It is known that expression of this gene is induced at low potassium concentrations. K^+^ is the most abundant cation in the cells and is necessary for growth, ribosome function, the activity of many enzymes and the maintenance of intracellular pH [37]. The genes of the *mrp* operon (multiple resistance and pH), encoding a multi subunit Na^+^/H^+^ antiporter (GYO_3450-3456, *mrpABCDEFG*), were also up-regulated after diterpene treatments. In our experiments, of the seven subunits, only the *mrpF* subunit was up-regulated at least 2X after all the treatments. The strongest activation was found in Sg3a samples where all subunits reached at least the 2X activation level. The *mrp* is a multifunctional locus that, in *Bacillus subtilis*, is involved in maintaining the Na^+^ and pH homeostasis and has a role in resistance to cholate. It has also been reported that the MrpF protein subunit alone can transport cholate and increase Na^+^ efflux [38]. The results suggest that the Na^+^/K^+^ homeostasis, and maybe the pH and the associated proton-motive force, are disturbed after diterpene treatments. Reversing these negative changes may be an important part of the response. Furthermore, it cannot be ruled out that MrpF exports molecules other than cholate, e.g., diterpenes, thus reducing their toxic effects. *YhbJ*, another putative efflux system component that may be involved in drug resistance and export, was also up-regulated after our treatments. 

Previously, it was established that the *yhbJ* co-regulated *yhb/yhc* genes were also activated after acidic stress, suggesting that these genes may also be involved in adaption to pH changes [39,40]. Activation of *yfiU*, a putative multidrug efflux transporter with unknown function, further confirms the importance of transmembrane transport during bacterial diterpene response.

Other genes sense and/or control the quality of membranes and membrane proteins. *liaH* and *liaI* are activated by oxidative stress, various cell wall antibiotics and antimicrobial peptides that can cause cell envelope stresses. LiaH is a homologue of the phage shock protein A and the IM30 protein of thylakoids. It binds and stabilizes distorted membrane regions. LiaI serves as the membrane anchor for LiaH [41]. Another membrane quality control gene is *htpX*, which encodes a stress-responsive membrane protease that controls the quality of membrane proteins. HtpX is a membrane-bound zinc metalloprotease involved in misfolded protein degradation after different stresses. Removing misfolded proteins from membranes promotes the survival of bacteria [42,43]. The up-regulation of various membrane quality control systems implies a robust and damaging effect of diterpenes on membranes. This assumption is supported by the general activation of the *yrhJ* (GYO_2953) gene in our samples. YrhJ is an NADPH-cytochrome P450 reductase that hydroxylates branched-chain fatty acids and has been implicated in altering cell membrane fluidity [44,45]. The key role of membrane composition in response to diterpenoids is further supported by the general up-regulation of glycerol-3-phosphate dehydrogenase (GYO_1218, *glpD*). GlpD is a key membrane enzyme involved in phospholipid biosynthesis. But, since it is also involved in carbohydrate metabolism and directly or indirectly in the cytoplasmic membrane electron transport chain as a quinone reductase, it can influence the stress response in several ways [46]. GlpD, through chorismate (shikimate pathway) and menaquinol synthesis, affects both the demethylmenaquinone and menaquinone synthesis. These latter molecules are components of the *B. subtilis* respiratory electron transport [47]. Moreover, up-regulation of other common genes of menaquinone synthesis (GYO_1191, *yhcB*; GYO_3335, *ytxM*; GYO_3490, *dhbC*) supports the importance of menaquinone dependent respiratory electron transport during diterpene-induced stress response. 

Finally, a putative glycosyltransferase (GYO_2344, *yojK*) was activated in all samples. Glycosyltransferases catalyze the transfer of sugar moieties to an acceptor molecule. *B. subtilis* has three putative UDP-glycosyltransferases with broad substrate ranges, capable of modifying various plant-derived antimicrobial molecules such as vanillin and quercetin [48]. Therefore, it is conceivable that these glycosyltransferases can reduce the harmful effect of diterpenes by modifying them.

The two commonly repressed genes also encode membrane proteins involved in ion and molecule transport (manganese uptake and C4-dicarboxylate transport). Down-regulation of these membrane transport processes in *B. spizizenii* may diminish the effect of diterpenes. The suppression of manganese uptake may be related to the oxidative stress response of the bacteria, but the role of the manganese in this response may be complex. On the one hand, Mn is a metal cofactor of Mn- superoxide dismutase (SOD) and acts as an antioxidant. On the other hand, Mn^2+^ can sensitize *B. subtilis* to H_2_O_2_ toxicity [49,50].

As mentioned above, ten common genes showed different patterns of up- or down-regulation depending on the treatments (Appendix A). A phosphate ABC transporter and permease (GYO_2763, *pstC*) involved in high-affinity phosphate uptake was activated in both 5h but repressed in both 1h samples, which suggests phosphate limitation and/or alkali stress during the longer incubation [51]. Four genes were up-regulated only in Sg3a-5h but down-regulated after other treatments. Three of them are membrane-localized proteins with different functions and an extracellular lipase. Three other genes were up-regulated only in Sg6-5h, of which two are sugar transporters. Finally, histidine degradation genes were repressed only in Sg3a-5h but activated after other treatments, implying that this treatment forces bacterial cells to use histidine as a source of carbon, energy and nitrogen [52].

The pairwise comparisons of treatments also showed moderate overlaps between the effects of the diterpenes (Table 4). If those genes were compared that were up-regulated after the same incubation period with different diterpenes, the number of common genes was relatively low (150 and 67 genes) and somewhat more in the case of down-regulated genes (194 and 82 genes). If the longer-incubated samples were compared, 121 oppositely regulated genes were detected (Sg3a-5h vs. Sg6-5h). In contrast, only one oppositely regulated common gene was found when the short-incubated samples were compared (Sg3a-1h vs. Sg6-1h). This gene (GYO_4330) is a wall-associated protein (*wapA*), which has a C-terminal toxic domain and can inhibit the growth of neighboring cells [53]. The differences in the number of oppositely regulated genes suggest that, during the shorter incubation, processes took place in the bacteria in roughly the same direction, whereas during the longer incubation period the processes could diverge.

When the results were compared between incubation periods (Sg3a-5h vs. Sg3a-1h and Sg6-5h vs. Sg6-1h), the number of genes up- or down-regulated in common was also moderate. The relatively high number of oppositely regulated genes (108 genes) suggests that Sg3a may trigger different bacterial responses in the early and late stages of the interaction (Table 4).

### 2.5. Enrichment Analyses Identified Significantly Up- or Down-Regulated Genes and Pathways Affected by Diterpene Treatments

Enrichment analyses reveal processes that are significantly altered in bacterial cells after treatments compared to control. While Gene Ontology (GO) annotates genes to biological processes, molecular functions and cellular components (https://www.geneontology.org, accessed on 16 December 2022), the Kyoto Encyclopedia of Genes and Genomes (KEGG, https://www.genome.jp/kegg, accessed on 16 December 2022) annotates genes to pathway level. The analyses were performed in two different ways: (i) calculated with significantly up- and down-regulated genes, (ii) calculated with significantly up- or down-regulated genes separately. The results of the analyses pointed to both common and specific changes, depending on the type and duration of the treatment. The molecular functions and biological processes of the GO enrichment analysis could separate the effect of shorter and longer treatments independently of the type of diterpene treatments. The shorter treatments affected similar processes, such as transport, oxidation–reduction processes and carboxylic acid transport (including amino acid transport). In contrast, the longer treatments, in addition to transport, mainly down-regulated several pathways, depending on the diterpene (Table 5 and Table 6).

### 2.6. Common Processes with Altered Gene Activities after Diterpene Treatments

Both GO and KEGG analyses showed that one of the most typical effects of diterpenes is interference with transport processes (Table 5, Table 6 and Appendix A and Figure 6). A substantial part of the transport changes is related to the membrane and related ion transport. The different treatments affected the transport processes to a different extent and in different ways. According to GO analyses, Sg6 has a somewhat milder effect on transport gene expression and this effect was mainly suppressive. 

Among the different transport activities, the KEGG analysis highlighted the activity changes of ABC-type transporters (ATP-binding cassette transporters) and showed that the most intense changes occurred in the Sg3a-5h samples with activating 35 and repressing 36 genes (Figure 6 and Appendix A). After this treatment, the ABC transporter genes of the osmoprotectant (choline), oligosaccharide, dipeptide and zinc/manganese/Iron (II) transporters were repressed, while the amino acid (cysteine, S-methylcysteine, arginine, D-methionine) and acetoin utilization transporters were activated (Appendix A). It was also observed that in the Sg3a-5h samples, typically, all subunits of the ABC transporters changed their activity, while only some subunits were activated or repressed after the other treatments. Some ABC transporters changed their activity oppositely after different treatments, depending on the diterpenes. For example, acetoin utilization-associated ABC transporter subunits (*ytrBCDEF*) were activated both after shorter and longer incubation with Sg3a (Sg3a-5h and Sg3a-1h) but were down-regulated in Sg6-5h. In the presence of an excess of carbohydrates, *B. subtilis* produces and secretes acetoin, which it then imports and reuses during the stationary phase and sporulation. Besides acetoin import, the ABC transporter YtrBCDEF may have a direct or indirect effect on the cell wall. It is also known that the transcription of this operon can be induced with various antibiotics. Overexpression of these genes in *B. subtilis* resulted in the formation of a thicker peptidoglycan layer and also affected biofilm formation [54,55]. Thus, activation of the ABC transporter *ytrBCDEF* in the Sg3a samples may be part of the cell wall stress responses. The reduced dipeptide transporter activity in the Sg3a-5h samples can be associated with decreased cell wall peptide uptake and peptidoglycan recycling [56].

Another general response is that the bacterial chemotaxis-related genes were repressed in all sample types. Except for Sg6-1h, the flagellar assembly genes were also strongly repressed (Figure 6 and Appendix A). Based on the literature data, the down-regulation of these processes appears to be a typical response of bacteria to various stresses [57,58] and can have both beneficial and detrimental consequences.

On the one hand, repression of these genes can block the ability of bacterial cells to respond to environmental signals by moving. On the other hand, by turning off unnecessary functions, the bacterium can concentrate on maintaining essential processes.

The other typical response is the changes in gene transcription related to the biosynthesis of secondary metabolites (Figure 6 and Appendix A). The strongest responses of these genes could be detected in the Sg6 samples, since the highest number of up- and down-regulated genes belong to this category. The Sg3a treatments induced weaker responses, since only the Sg3a-1h samples showed significant gene activation. The activated secondary metabolite-related genes are involved in various pathways (e.g., isoprenoids, menaquinone, arginine, aromatic amino acids, heme, lysine, threonine, branched-chain amino acids, leucine, methionine and threonine biosynthesis) directly or indirectly connected to secondary metabolite production.

### 2.7. Treatment Specific Up-Regulated Processes Enriched after Diterpene Treatments

The GO enrichment analysis resulted in relatively few significantly up-regulated bacterial GO terms (Table 5 and Table 6). One was the oxidoreductase activity/oxidation–reduction processes, which was more characteristic after shorter treatments (Sg3a-1h and Sg6-1h). Genes with oxidoreductase activity are involved in a wide range of bacterial metabolism, including carbohydrates, energy, lipids, amino acids, cofactors, vitamins, etc. The up-regulation of these genes in the 1h samples suggests that the treatments induce more intense metabolic activity than controls, to adapt to diterpene treatments.

After the Sg6-5h treatment, two overlapping terms, “developmental” and “sporulation”, were enriched significantly. The developmental term includes germination genes in addition to sporulation genes. Among the sporulation genes, the *spo0A* gene (GYO_2679), which regulates sporulation transcription, was also activated. Generally, the initiation of sporulation of *Bacillus* occurs in harsh environments that are suboptimal for growth, such as nutrient depletion [59]. After the diterpene treatments, compared to the control samples, bacterial growth is slower in the LB medium. Therefore, it cannot be assumed that a faster nutrient depletion initiated the sporulation process in the treated samples. On the other hand, it is conceivable that some nutrients are depleted much faster in diterpene-treated samples than in the control samples, or some other unfavorable changes for the bacteria have activated the genes related to sporulation. The role of parallel activation of spore germination genes with sporulation genes is unclear and requires further investigation.

Besides sporulation, GO analysis showed that carbohydrate metabolism-related processes, such as hydrolase activity on glycosyl bonds and carbohydrate transmembrane transporter activity, were enhanced in the Sg6-5h samples. The higher activity of genes involved in the hydrolyzation of different carbohydrates and their transport across the membranes implies more intensive carbohydrate metabolism after Sg6. This response may not be a general reaction after diterpene treatments because hydrolase activity genes were down-regulated in the Sg3a-5h samples.

The KEGG enrichment result pointed to further up-regulated pathways. In the Sg3a-5h samples, the up-regulation of the teichoic acid biosynthesis reveals that bacterial cells respond to the diterpene treatments with cell wall modification. Wall teichoic acids of Gram-positive bacteria affect cell wall integrity, influence cell division and cell morphology, scaffold some cell wall assembly proteins, exclude other proteins and influence protein activity through ion chelation. Wall teichoic acids-tailoring modifications are involved in biofilm formation, virulence and antimicrobial resistance [60].

After Sg6 treatment, several amino acid biosynthesis/metabolism related pathways were up-regulated significantly. Those of valine, leucine, isoleucine and tryptophan were found to be significant in the shorter treatment (Sg6-1h). In contrast, strong up-regulation of arginine biosynthesis genes was observed after both shorter and longer Sg6 treatments. The metabolism of arginine is linked to various pathways that contribute to the bacterial cell’s ability to adapt and survive under stress conditions. For example: (i) Arginine is a precursor for nitric oxide (NO) synthesis. NO is a signaling molecule that plays a role in the response to various stresses [61,62]. (ii) *Bacillus subtilis* utilizes arginine as a precursor for polyamine synthesis, including putrescine and spermidine. Among other things, polyamines are involved in adaptation to stress, protection of bacterial cells and biofilm formation [63,64]. Therefore, the increase in arginine synthesis in the Sg6 treatment can be part of the stress response of the bacteria in different ways.

### 2.8. Treatment Specific Down-Regulated Processes Enriched after Diterpene Treatments

It is very likely that suppressed pathways and processes are responsible for bacterial dysfunction by diterpenes. The pathways significantly inhibited by the two diterpenes were divergent (Table 5, Table 6 and Appendix A and Figure 6).

In the Sg6-5h samples, the enrichment results showed repression of a wide range of the processes involving organonitrogen compounds (organonitrogen compound metabolic process: 127 genes) (Table 5 and Table 6). Within this category, one of the most characteristic changes was the repression of protein synthesis-related genes and processes. Both molecular functions and biological process terms, such as translation, structural constituent of ribosome and catalytic activity acting on a tRNA, contained several down-regulated genes. These genes participate in ribosome assembly, transfer RNA biogenesis, etc. The KEGG pathway enrichment analysis also supported the down-regulation of ribosome-related genes in this sample (Figure 6).

The carbohydrate metabolic process and related energy metabolism was also reduced after longer treatments (5h samples). Moreover, in the Sg6-5h samples, the ATP biosynthetic process (oxidative phosphorylation and ATP synthase genes) was down-regulated (Table 6 and Figure 6). Simultaneously, starch and sucrose metabolism genes, including the oligosaccharide internalizing phosphotransferase system (PTS), were activated (see also the up-regulation of carbohydrate metabolism-related processes in Section 2.7). These changes were probably activated to compensate for the decrease in energy production (ATP synthesis).

In the Sg6-5h samples, the purine ribonucleoside triphosphate binding GO category, which affects wide range of processes, was another significantly down-regulated term (94 genes). This term includes repressed GTP binding proteins which involve important regulation reactions (16 genes). Some of them participate in ribosome assembly (*cpgA*, *obg*, *engA*, *rbgA*) or translation initiation (*tufA*, *ylaG*), and down-regulation of these genes coincides with translation repression mentioned above. 

Changes in the biosynthesis and degradation of lipids and fatty acids are probably closely related to the membrane perturbation induced by diterpenes (Table 6, Figure 6 and Appendix A). Interestingly, opposite processes could be observed after the two diterpene treatments after longer incubations. While in the Sg6-5h samples, the expressions of fatty acid and lipid biosynthetic process genes were decreased and the fatty acid degradation genes were up-regulated, the trend was opposite in the Sg3a-5h samples. Shorter treatments showed mixed trends, as Sg3a treatment (Sg3a-1h) induced both up- and down-regulation of the transcription of fatty acid synthesis genes and activation of degradation genes, while short Sg6 treatment (Sg6-1h) induced moderate up-regulation of some fatty acid synthesis and degradation genes. This result implies that the Sg6 treatment caused stronger and longer lasting effects on cell membranes and fatty acid metabolism. It seems that, after Sg3a, the bacteria could recover to a certain extent from the membrane stress by inducing the fatty acid synthesis and repressing the fatty acid degradation process.

The GO enrichment results implied significant repression of the signal transduction-related genes (Table 6 and Appendix A), especially after longer treatment with Sg3a (Sg3a-5h). Most of these signal-related genes function in different two-component systems. Two-component systems usually contain a membrane-bound histidine kinase that senses a specific environmental stimulus and a matching response regulator. Through these systems, the bacterial cells are able to mediate cellular responses, typically influencing the expression of target genes. Several two-component genes related to the flagellar assembly and chemotaxis regulation were down-regulated, which may be associated with the general repression of chemotaxis and flagellar assembly processes mentioned above. Other repressed two-component system genes are involved in lantibiotic biosynthesis and sporulation.

Another important element of the bacterial response to stress is changes in the expression of transcription factors (TF). The results of GO enrichment showed that the shorter treatment of Sg6 (Sg6-1h) induced significant repression of several TFs compared to controls. Down-regulation of TFs, depending on whether they are activators or repressors, leads to up- or down-regulation of the genes and processes they regulate. Repressed TFs affect different aspects of bacterial stress responses. Some of them regulate bacterial metabolism, such as the activation of cysteine biosynthesis (GYO_4151, *cysL*) or repression of carbohydrate utilization (GYO_3275, *melR*, melibiose and raffinose; GYO_4408, *gntR*, gluconate). Down-regulation of the *fur* gene (GYO_2588), which acts as a transcriptional repressor of iron uptake, indicates limited iron accessibility after diterpene treatment. Other TFs may be involved in the activation of resistance responses against oxidative and electrophile stress (GYO_1695, *mhqR,* transcriptional repressor). However, the complexity of the regulation is indicated by the fact that another gene contributing to the hydrogen peroxide resistance of *Bacillus subtilis*, an ECF-type sigma factor (*ylaC*, GYO_1815), was repressed. In addition, GYO_2949, a SigV RNA polymerase sigma factor required for resistance to lytic enzymes, was also down-regulated. The latter two changes due to diterpene treatments may mean transcriptional processes that can negatively influence the adaptation of the bacterial cell to the harmful effects.

After both short treatments (Sg3a-1h, Sg6-1h) the carboxylic acid transmembrane transporter activity was significantly repressed. Carboxylic acid transport, including amino acid transport, may influence a variety of metabolic pathways. One of the down-regulated carboxylic acid transporters is the GYO_0666 (*dctP*), which is responsible for uptake of succinate, fumarate, malate and oxaloacetate. As these molecules are TCA cycle (tricarboxylic acid cycle) intermediates, their levels can affect energy production and synthesis of other metabolites. Other down-regulated transporters are amino acid transporters such as glutamate and aspartate (GYO_1320, *gltT*), glutamine (GYO_2182, *alsT*), lysine (GYO_3644, *yvsH*), serine/threonine (GYO_1603, *stet*) and isoleucine and valine (GYO_3206, *braB*). On the one hand, a decrease in the availability of these amino acids can directly negatively affect protein synthesis. On the other hand, since these amino acids can participate in many metabolic pathways, including the synthesis of other amino acids, they can also indirectly affect protein synthesis.

In the Sg3a-1h samples, the GO enrichment pointed out the repression of genes encoding several heme-binding proteins. Some of them are catalases (GYO_1168, *katA*; GYO_4264 *katX*; GYO_4310, *katE*), all of which act against oxidative stress by degrading hydrogen peroxide. Therefore, the down-regulation of these catalases may lead to a decrease in resistance to oxidative stress induced by diterpene treatment, while other repressed heme-binding genes encode cytochrome c oxidase subunits of oxidative phosphorylation (GYO_2481, GYO_1834, GYO_1833) and thus likely negatively affect energy metabolism.

### 2.9. Comparison of Effect of Diterpene Treatments on Transcriptome with Changes Induced under Different Environmental and Nutritional Conditions

In order to better understand the changes in the *B. spizizenii* transcriptome as a result of treatment with diterpenes, it may be helpful to compare our results with those obtained under various environmental and nutritional conditions. Nicolas P. et al. [65] measured the gene expression changes in *B. subtilis* under several conditions covering various nutrients, aerobic and anaerobic growth, the development of motility, biofilm formation, adaptation to diverse stresses, high cell density fermentation, development of competence for genetic transformation, spore formation and germination. Another paper [66] using these data transformed the high dimensional transcriptomics data into a two-dimensional map. In this way, 10 clusters were identified, representing different changes in the transcriptome. Four clusters mainly contained samples grown under similar conditions, and six were mixed clusters containing samples grown under other conditions. They also performed GO pathway enrichment analyses to find bacterial processes enriched in the clusters. To find treatments similar to diterpenes, the GO enrichment results of these *B. subtilis* clusters and our *B. spizizenii* treatments were compared (Appendix A). Some clusters showed little overlap with our GO categories. Interestingly, despite bacterial growth inhibition, little similarity could be observed between the diterpene and mitomycin antibiotic treatments (cluster 8, including antibiotic treatment lasting 90 min and cold treatment). Diterpene treatments also showed limited overlap with clusters 5 and 9, which contain some stresses such as salt, ethanol and heat in cluster 9, and heat, salt, low-phosphate treatment, biofilm formations and various carbon source changes in cluster 5.

The highest overlap with other conditions could be seen in the case of 5h treatments, especially after Sg6. This observation also confirms that the greatest effect was induced by the longer incubation with Sg6. This treatment has commonly enriched processes with all clusters except the above-mentioned three clusters (5, 8 and 9). Shorter incubation (Sg3a-1h and Sg6-1h) resulted in much less overlapping processes.

The Sg6-5h treatment shared enriched down-regulated GO terms with clusters 1, 4 and 6, such as translation and ribose phosphate biosynthetic processes including ATP biosynthesis. These clusters include diverse conditions and treatments, such as early-stage sporulation, transfer from LB medium to minimal medium (90 min competence), transition and stationary phase in LB and LB+glucose medium, individual/confluent colonies on LB agar, swarming cells from 1% agar LB plates, 40 min after mitomycin treatment, carbon supplement from malate to malate+glucose in M9 minimal medium, stationary phase in M9+glucose medium, transfer from high phosphate-defined medium to limited phosphate-defined medium in stationary phase, and glucose-exhausted M9 medium (starvation in stationary phase). Remarkably, most of these changes could be induced during starvation and the late growing phase of the bacteria, which suggests that the Sg6 treatment triggers changes that simulate the late phase of growth and induce starvation-like conditions in *B. spizizenii*.

Oxidative stress (cluster 7), induced by different agents (diamide, paraquat and H_2_O_2_) in *B. subtilis* also showed notable overlap with Sg6 treatment. Oxidative stresses and longer incubation with Sg6 (Sg6-5h) had common down-regulated GO terms such as ATP biosynthetic process, cell motility, carbohydrate derivative biosynthetic process and cation transmembrane transport. In contrast, after short-duration Sg6 treatment (Sg6-1h), the genes of transcriptional regulatory-related terms showed opposite regulation compared to oxidative stress (down-regulated in Sg6-1h and up-regulated during oxidative stress).

Other notable overlaps could be observed with spore germination processes. Common enriched repressed processes were “flagellum-dependent cell motility”, “chemotaxis” and “carbohydrate metabolic process”. However, the translation and transcription regulation-related processes and purine ribonucleotide binding implied opposite directions because these types of enriched genes were up-regulated during germination but repressed after Sg6 treatments.

## 3. Materials and Methods

### 3.1. Bacterial Strains and Microdilution Assays

*Bacillus spizizenii* (*B. spizizenii*, DSM 618, Merck, Darmstadt, Darmstadt, Germany) and *Bacillus subtilis* (*B. subtilis* strain F1276, Central Food Research Institute, Budapest, Hungary) were grown overnight on solid LB medium (10 g/L tryptone, 5 g/L yeast extract and 10 g/L sodium chloride, 12 g/L agar) at 37 °C. *Rhodococcus fascians* (*R. fascians* NCAIM B.01608, National Collection of Agricultural and Industrial Microorganisms, Budapest, Hungary) was grown overnight on solid Waksman’s medium (5 g/L peptone, 5 g/L meat extract, 5 g/L sodium chloride, 10 g/L glucose, 12 g/L agar, pH adjusted to 7.2 with a 40% aqueous sodium hydroxide solution) at 26 °C. The bacterial cells were washed off the agar surface, suspended in liquid LB, and the suspensions were adjusted with LB to OD_600_ = 0.2 (10^8^ cells/mL). To obtain a 10^5^ cell/mL suspension, this was diluted thousand times in liquid LB. 

Clerodane diterpenes were isolated from *Solidago gigantea* roots, as previously described [12]. The minimal bactericidal concentration (MBC), minimal inhibitory concentration (MIC) and half-maximal inhibitory concentration (IC_50_) of the isolated compounds (stock: 2.5 mg/mL in ethanol) and, as a positive control, gentamicin (100 µg/mL in ethanol, Merck KGaA, Darmstadt, Germany) were determined using non-treated flat-bottom 96-well microplates (VWR, cat. no. 734-2781). Ethanol served as the negative control. A two-fold dilution series starting from 5 µL of each isolate was prepared in ethanol in the wells. When the ethanol evaporated under a sterile laminar airflow, 150 µL of a bacterial suspension (10^5^ cells/mL) was added to each well. Starting values of the absorbance at 600 nm were measured using a microplate spectrophotometer (Labsystems Multiscan MS 4.0, Thermo Fisher Scientific, Waltham, MA, USA). The bacterial cultures were shaken at 900 rpm (at 37 °C for *B. subtilis* and at 26 °C for *R. fascians*) with a PHMP Twin Microplate Shaker-Incubator Thermoshaker (Grant Inc., Beaver Falls, PA, USA) for 24 h. At the end of the incubation period (20 h for *B. subtilis* and 72 h for *R. fascians*), the OD_600_ values were measured again. The experiment was repeated twice with three parallels, and the results were averaged. The MBC was determined by putting 5 µL suspensions from each well onto appropriate agar layers to monitor the presence or absence of bacterial colonies.

### 3.2. Treatment of B. spizizenii with Diterpenes Sg3a and Sg6 for RNA Purification

*B. spizizenii* was cultured overnight on LB agar at 37 °C. The bacterial cells taken from the agar were suspended in liquid LB and these suspensions were adjusted to OD_600_ = 0.2 (10^8^ cell/mL) or OD_600_ = 0.5 (6 × 10^8^/mL) in LB. The purified diterpenes Sg3a and Sg6 (dissolved in ethanol) were added at time 0 to bacterial suspensions (7.5 mL in a 100 mL size Erlenmeyer flask) at a final concentration of 3.5 μg/mL and 1.25 μg/mL, respectively. In the control treatments, ethanol was added to bacterial suspensions in the same volumes as the diterpenes. The bacterial suspensions that were started at OD_600_ = 0.5 were further incubated for one hour, and, for those at OD_600_ = 0.2, the treatments took five hours. The bacterial cultures were shaken at 120 rpm at 37 °C. Following incubation, 2 mL bacterial suspensions were centrifuged at 16,000× *g* for 2 min at 4 °C and the supernatant was discarded. The bacterial pellets were frozen in liquid nitrogen and stored at −70 °C. Samples were collected for four biological repeats.

### 3.3. RNA Purification from B. spizizenii 

For RNA purification, the frozen bacterial pellets were dissolved in 200 µL 10 mM Tris-EDTA (TE) buffer pH 8.0 containing lysozyme (3 mg/mL, Merck KGaA, Darmstadt, Germany), and incubated at 25 °C for 10 min. Then, 500 µL RNAzol RT (Merck KGaA, Darmstadt, Germany) was added, mixed by vortexing and incubated for 15 min at room temperature. After incubation, the mixture was centrifuged at 12,000× *g* for 15 min at 4 °C. An amount of 400 µL supernatant was removed and mixed with 400 µL ethanol (Reanal, Budapest, Hungary) in a new 1.5 mL tube. The RNA from the mix was further purified with a Direct-zol RNA MiniPrep Plus Kit (Zymo Research, Irvine, CA, USA), with DNase I treatment on the column. RNA was eluted with 50 µL DNase/RNase-free water and stored at −70 °C. The RNA quantities and the qualities were checked with a NanoDrop ND-1000 (Thermo Fisher Scientific, Waltham, MA, USA) and by a native 1% agarose gel.

### 3.4. RNA Sequencing, Data Processing and Analysis

The RNA sequencing reactions were performed by Novogene (Nanjing, China). Before RNA sequencing, the RNA qualities were checked with an Agilent 5400 (Agilent, Santa Clara, CA, USA). The RNA library preparations (rRNA removal, RNA fragmentation, cDNA reverse transcription, stranded RNA library preparation) used only those RNA samples whose quality reached the criteria of the RNA-Seq (≥6.0 RNA Integrity Number). After library preparation, paired-end 150 bp sequencing (Illumina NovaSeq 6000 platform, San Diego, CA, USA) was used to sequence the samples. Before bioinformatics analysis, the resulting data went through quality data control. Raw data (raw reads) of FASTQ format were first processed through fastp [67]. In this step, clean data (clean reads) were obtained by trimming reads containing adapter and removing poly-N sequences and low-quality reads. At the same time, Q20, Q30 and GC content of the clean data were calculated. All the downstream analyses were based on the clean and high-quality data. The reference genome (*Bacillus spizizenii* TU-B-10, GenBank assembly accession: GCA_000227465.1) and gene model annotation files were downloaded from the genome website directly (https://bacteria.ensembl.org, accessed on 16 December 2022). Bowtie2 was deployed for both building the index of the reference genome and aligning the clean reads to it [68]. The gene expression level was measured by featureCounts [69] and measured in Fragments Per Kilobase of transcript sequence per Millions base pair sequenced (FPKM) [70]. Differential expression analysis was performed using the DESeq2 R package [71]. The resulting *p*-values were adjusted using Benjamini and Hochberg’s approach for controlling the false discovery rate [72]. Genes with an adjusted *p*-value < 0.05 found by DESeq2 were assigned as differentially expressed. The ClusterProfiler R package [73] was used for both Gene ontology (GO) enrichment and Kyoto Encyclopedia of Genes and Genomes (KEGG, [74]) pathway analysis of the differentially expressed genes (DEGs). Terms with corrected *p* values of less than 0.05 were considered significantly enriched by differentially expressed genes. 

## 4. Conclusions

Gram-positive bacteria are generally more sensitive to terpenes than Gram-negatives. This difference probably originates from peculiarities of cell wall composition [75]. Our previous study supported this observation with clerodane-type diterpenes isolated from giant goldenrod roots [12]. In the present study, we broadened the investigation of the effect of these diterpenes on diverse Gram-positive bacteria. Among the seven compounds, Sg3a (a dialdehyde) and Sg6 (solidagoic acid B) showed the strongest antimicrobial activity. For both *Bacillus* strains, the MIC and MBC values of these diterpenes were the same, confirming the strong bactericidal activity of these compounds. The outstanding antibacterial activity of Sg3a and Sg6 may result from their α, β-unsaturated carbonyl part (conjugated C=C and C=O double bonds), a highly reactive site due to electrophilic carbonyl- and β-carbons. α, β-unsaturated carbonyls could target, e.g., multidrug efflux pumps of bacteria, diminishing their antibiotic resistance [76]. Our genome-wide transcriptomic analysis provided additional data on how diterpenes might act on bacteria. It highlighted common and specific changes caused by Sg3a and Sg6 during incubation. The results also indicated that Sg6 exerted its effect slower than Sg3a, but ultimately its impact was higher. The activated common genes and some specifically altered gene activities suggest that one of the primary targets in the bacteria is the cell envelope. Some of these genes are well-known envelope stress response genes. Diterpene-induced gene activity changes related to biosynthesis, degradation and modification of lipids and fatty acids have also been associated with membrane perturbations. However, the high complexity of the reactions was indicated by the fact that, besides shared membrane composition-related responses, there were also substance-specific alterations in opposite directions. Another typical response was the transcriptional modification of membrane transport genes. Its purpose may be to restore the cell’s homeostasis (e.g., that of K^+^, Na^+^ and pH) and/or to remove harmful molecules. Down-regulated genes underscored several suppressed pathways likely involved in bacterial inhibition. However, some repressed processes may also be part of adaptation responses. One such change was the general suppression of bacterial chemotaxis and flagellar assembly genes, which can block the ability of bacterial cells to respond to environmental signals by movement but allow the bacterium to focus on maintaining essential metabolism. Suppression of genes related to protein synthesis in Sg6-5h may be another example of a possible dual effect: it impedes many bacterial activities but may also serve to redistribute resources. Down-regulation of genes for other key processes, such as oxidative phosphorylation and ATP synthesis, signal transduction and transcription factors, was also detected. Part of the activated pathways is related to bacterial cell wall modification, e.g., teichoic acid biosynthesis. Others, such as oxidation–reduction, amino acid metabolism and secondary metabolites, may be the result of the more intense metabolic activity of adaptation to diterpene treatments. A comparison of the effect of diterpenes with those induced under different environmental and nutritional conditions revealed several overlapping responses. For instance, diterpene treatments, especially the longer one with Sg6 (Sg6-5h), showed notable overlap with oxidative stress and starvation-like conditions, as reflected in the enrichment of sporulation genes.

## Figures and Tables

**Figure 1 ijms-25-01531-f001:**
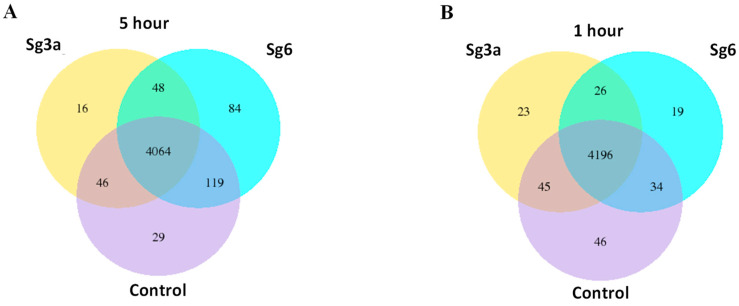
Number of transcripts detected in *B. spizizenii* in control and diterpene (Sg3a and Sg6) treated samples. RNA was isolated (**A**) after 5 h or (**B**) after 1 h of incubation.

**Figure 2 ijms-25-01531-f002:**
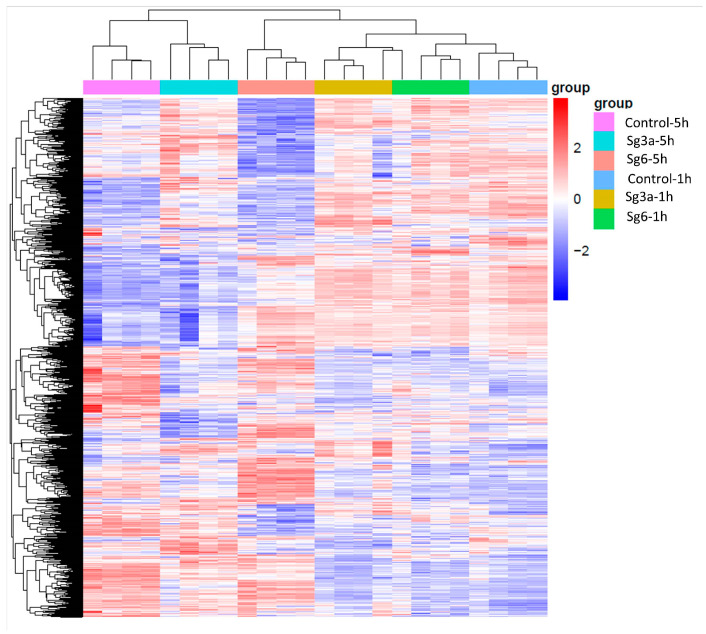
Results of FPKM (Fragments per kilobase of transcript sequence per millions base pair sequenced) cluster analysis, which clustered using the log_10_(FPKM+1) value. Red and blue colors indicate genes with high and low expression levels, respectively. Samples obtained after diterpene (Sg3a and Sg6) treatments or in control samples. *B. spizizenii* RNA was isolated after 5 h or after 1 h of incubation.

**Figure 3 ijms-25-01531-f003:**
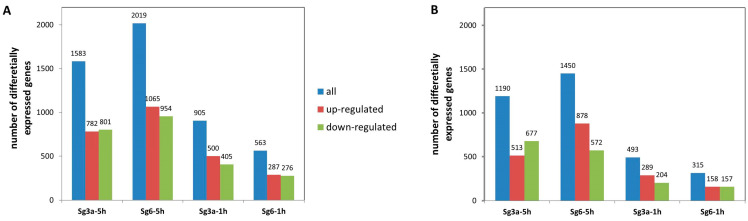
Number of genes significantly up- or down-regulated after diterpene (Sg3a and Sg6) treatments compared to control. (**A**) Number of all genes significantly up- or down-regulated (|log_2_(FoldChange)| > 0, adjusted *p*-value < 0.05). (**B**) Number of genes significantly up- or down-regulated at least 2X (|log_2_(FoldChange)| > 1, adjusted *p*-value < 0.05). *B. spizizenii* RNA was isolated after 5 h or after 1 h of incubation.

**Figure 4 ijms-25-01531-f004:**
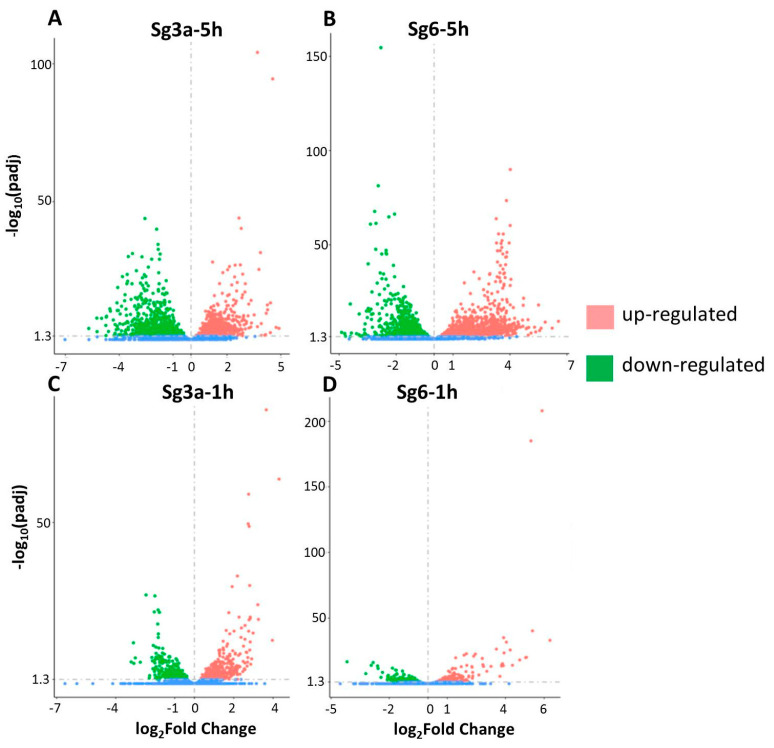
Distribution of differentially expressed genes (|log_2_(FoldChange)| > 0, *p*-value < 0.05). Diterpene (Sg3a and Sg6) treatments compared to controls. *B. spizizenii* RNA was isolated after 5 h or after 1 h of incubation. (**A**) Sg3a-5h vs. control-5h. (**B**) Sg6-5h vs. control-5h. (**C**) Sg3a-1h vs. control-1h. (**D**) Sg6-1h vs. control-1h. The Y-axes differ in scales.

**Figure 5 ijms-25-01531-f005:**
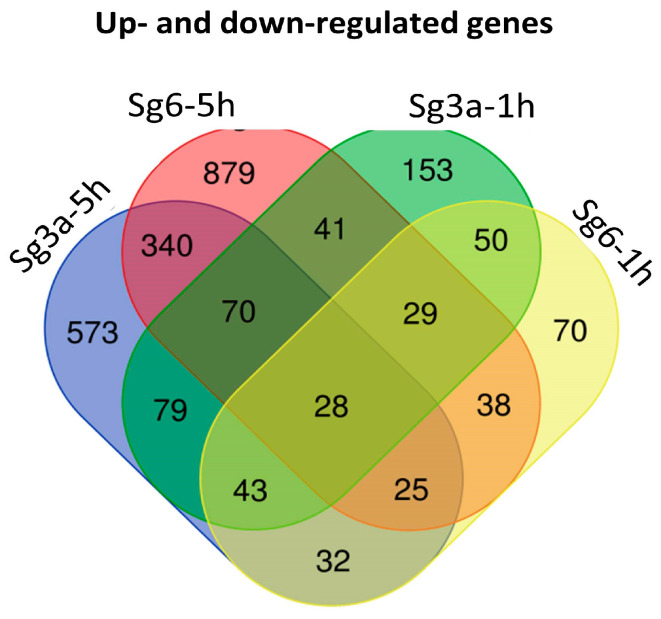
Number of common and specific genes significantly up- or down-regulated after diterpene (Sg3a or Sg6) treatments compared to controls (|log_2_(FoldChange)| > 1, adjusted *p*-value < 0.05). *B. spizizenii* RNA was isolated after 5 h or after 1 h of incubation. https://bioinformatics.psb.ugent.be/webtools/Venn/, accessed on 1 September 2023.

**Figure 6 ijms-25-01531-f006:**
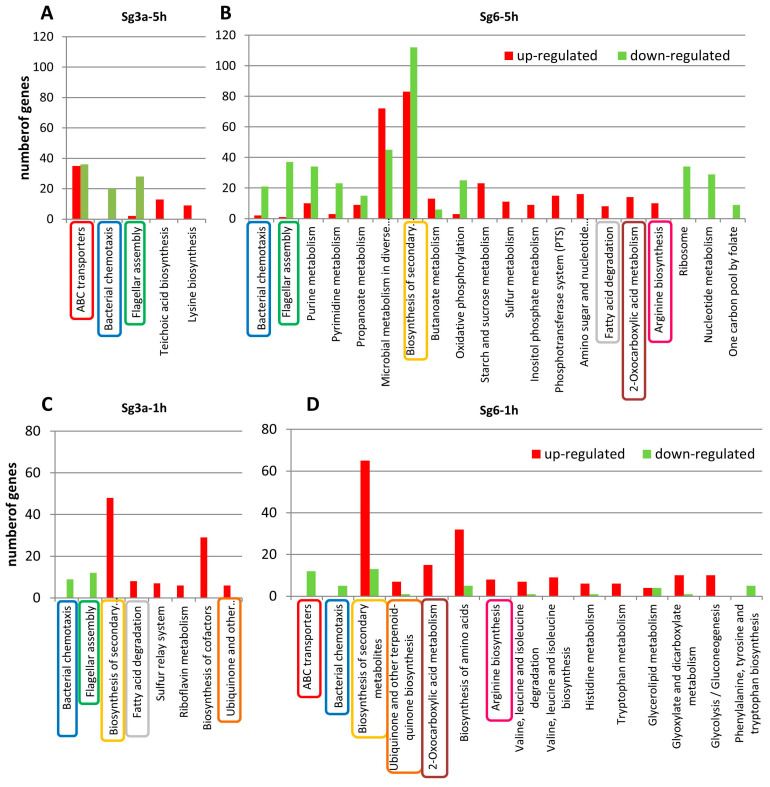
KEGG pathways enrichment results of diterpene treatments. Diterpene (Sg3a and Sg6) treated *B. spizizenii* samples were compared to control (Cont) samples. (**A**) Sg3a-5h vs. control-5h. (**B**) Sg6-5h vs. control-5h. (**C**) Sg3a-1h vs. control-1h. (**D**) Sg6-1h vs. control-1h. The numbers of genes up- or down-regulated in the significantly enriched pathways are marked in red and green, respectively. KEGG terms with padj < 0.05 were regarded as significant. The common pathways in the different treatments were marked with a rectangle framed in the same color.

**Table 1 ijms-25-01531-t001:** Half-maximal inhibitory concentration (IC_50_ values, μg/mL) for clerodane diterpenes.

	Sg1 ^a^	Sg2	Sg3a	Sg3b	Sg4	Sg5	Sg6	Gent ^b^
*B. subtilis*								
mean	23.74	44.15	4.39	- ^e^	7.89	56.67	2.07	0.77
SD ^c^	1.42	17.63	0.44	-	0.62	22.76	0.07	0.14
RSD% ^d^	5.97	39.92	10.01	-	7.88	40.16	3.21	18.50
*B. spizizenii*								
mean	16.36	36.00	4.49	-	8.48	32.03	1.65	0.79
SD	0.61	0.80	0.13	-	0.21	3.67	0.14	0.15
RSD%	3.75	2.23	2.90	-	2.51	11.46	8.43	18.66
*R. fascians*								
mean	8.32	15.83	1.59	7.09	4.01	1.93	1.43	3.58
SD	0.27	1.22	0.07	0.14	0.04	0.25	0.22	0.20
RSD%	3.25	7.72	4.65	1.98	0.98	12.87	15.43	5.52

^a^ Clerodane type diterpenes (Sg1, Sg2, Sg3a, Sg3b, Sg3c, Sg4, Sg5, Sg6) were isolated from giant goldenrod root. ^b^ Gent: gentamicin positive control. ^c^ SD: standard deviation. ^d^ RSD%: relative standard deviation. ^e^ -: Not determined.

**Table 2 ijms-25-01531-t002:** Minimal inhibitory concentration (MIC) and minimal bactericidal concentration (MBC) values (μg/mL) for clerodane diterpenes.

		Sg1 ^a^	Sg2	Sg3a	Sg3b	Sg4	Sg5	Sg6	Gent ^b^
*B. subtilis*	MIC	41.66	83.33	5.2	- ^c^	10.41	-	2.6	1.04
	MBC	41.66	83.33	5.2	-	41.66	-	2.6	1.66
*B. spizizenii*	MIC	41.66	83.33	8.3	-	10.4	-	2.08	1.66
	MBC	41.66	83.33	8.3	-	20.83	-	2.08	1.66
*R. fascians*	MIC	10.4	20.8	4.1	16.7	5.2	4.1	2.6	8.3
	MBC	41.7	41.7	8.3	-	10.4	-	8.3	16.7

^a^ Clerodane-type diterpenes (Sg1, Sg2, Sg3a, Sg3b, Sg3c, Sg4, Sg5, Sg6) were isolated from giant goldenrod root. ^b^ Gent: gentamicin positive control (100 µg/mL). ^c^ -: Not determined.

**Table 3 ijms-25-01531-t003:** Common differentially expressed genes after diterpene (Sg3a or Sg6) treatments (1 or 5 h).

Gene Id ^a^	Sg3a-5h	Sg6-5h	Sg3a-1h	Sg6-1h	Similarity, Function
GYO_0650	1.6 ^b^	1.2	2.2	1.9	*B. subtilis* (BSU_04320) *ydaO*/*kimA*, high-affinity K^+^/H^+^ symporter
GYO_1109	2.9	1.5	4.3	1.4	*B. subtilis* (BSU_08400) *yfiU*, similar to multidrug-efflux transporter
GYO_1188	3.2	2.5	3.0	2.8	*B. subtilis* (BSU_08990) *yhbI* similar to transcriptional regulator
GYO_1189	2.5	2.2	2.8	4.0	*B. subtilis* (BSU_09000) *yhbJ*, putative efflux system component
GYO_1190	2.8	4.0	2.8	3.9	*B. subtilis* (BSU_09010) *yhcA*, c-di-AMP exporter
GYO_1191	2.5	3.6	2.8	3.9	*B. subtilis* (BSU_09020) *yhcB* similar to oxidoreductase
GYO_1192	2.7	3.6	2.6	3.5	*B. subtilis* (BSU_09030) *yhcC* unknown, hypothetical membrane protein
GYO_1218	1.2	1.1	1.3	2.1	*B. subtilis* (BSU_09300) *glpD*, glycerol-3-phosphate dehydrogenase
GYO_1674	2.3	1.8	2.8	1.2	*B. subtilis* (BSU_13490) *htpX*, stress-responsive membrane protease
GYO_2344	2.4	2.5	2.6	2.9	*B. subtilis* (BSU_19420) *yojK* similar to macrolide glycosyltransferase
GYO_2953	1.1	1.0	2.1	1.7	*B. subtilis* (BSU_27160) *yrhJ*, cytochrome P450, fatty acid metabolism
GYO_3335	1.1	2.1	1.0	1.7	*B. subtilis* (BSU_30810) *ytxM*, ubiquinone and menaquinone biosynthesis
GYO_3455	1.2	1.4	2.3	1.1	*B. subtilis* (BSU_31650) *mrpF*, Na+/H+ antiporter subunit, sodium export
GYO_3490	2.3	2.0	1.0	1.0	*B. subtilis* (BSU_31990) *dhbC*, isochorismate synthase
GYO_3618	1.6	2.3	2.9	2.8	*B. subtilis* (BSU_33120) *liaH*, Two-component system, accessory subunit of the TatAY-TatCY protein secretion complex, resistance against oxidative stress and cell wall antibiotics, protein secretion
GYO_3619	1.3	1.6	2.9	2.3	*B. subtilis* (BSU_33130) *liaI*, Two-component system, membrane anchor for LiaH
GYO_0666	−1.7	−1.2	−1.3	−1.3	*B. subtilis* (BSU_04470) *dctP*, C4-dicarboxylate transport protein, Two-component system, uptake of succinate, fumarate, malate and oxaloacetate via proton symport
GYO_3331	−2.0	−1.2	−1.8	−1.8	*B. subtilis* (BSU_30770) *mntA*, manganese ABC transporter (binding protein, lipoprotein), manganese uptake

^a^ KEGG entry of *B. spizizenii* genes. ^b^ log_2_ FoldChange.

**Table 4 ijms-25-01531-t004:** Number of common and specific genes significantly up- or down-regulated after 1 h or after 5 h of diterpene treatments (Sg3a or Sg6) compared to control.

Treatments	Common Up-Regulated Genes	Common Down-Regulated Genes	Oppositely Regulated Genes
Sg3a-5h vs. Sg6-5h	150	194	121
Sg3a-1h vs. Sg6-1h	67	82	1
Sg3a-5h vs. Sg3a-1h	83	29	108
Sg6-5h vs. Sg6-1h	56	32	32

**Table 5 ijms-25-01531-t005:** Simplified results of GO Molecular function enrichment analysis. Significantly enriched GO terms obtained after 1 h or after 5 h of diterpene treatments (Sg3a or Sg6) compared to control.

GO Term ^a^		5 h		1 h	
		Sg3a	Sg6	Sg3a	Sg6
GO:0005215	transporter activity	up/down ^b^ 58/51	down 18	up/down 24/46	up/down 14/30
GO:0016491	oxidoreductase activity	up/down 3/7		up 66	up/down 60/4
GO:0016798	hydrolase activity, acting on glycosyl bonds	down 16	up 17		
GO:0015144	carbohydrate transmembrane transporter activity		up 16		
GO:0020037	heme binding	down 16			
GO:0003723	RNA binding		down 39		
GO:0003735	structural constituent of ribosome		down 27		
GO:0140101	catalytic activity, acting on a tRNA		down 20		
GO:0035639	purine ribonucleoside triphosphate binding (including GTP binding)		down 94		
GO:0046943	carboxylic acid transmembrane transporter activity			down 8	down 6
GO:0140110	transcription regulator activity				down 14

^a^ GO terms with padj < 0.05 were regarded as significant. ^b^ Number of up- or down-regulated genes associated with the GO term. Gene numbers in red, green and yellow indicate that the GO term contains, up-, down-, and up-and-down-regulated genes, respectively.

**Table 6 ijms-25-01531-t006:** Simplified results of GO Biological process enrichment analysis. Significantly enriched GO terms obtained after 1 h or after 5 h of diterpene treatments (Sg3a or Sg6) compared to control.

GO Term ^a^		5 h		1 h	
		Sg3a	Sg6	Sg3a	Sg6
GO:0006810	transport	up/down ^b^ 79/90		down 57	down 37
GO:0055114	oxidation–reduction process			up 55	up 47
GO:0032502	developmental process		up 23		
GO:0043934	sporulation		up 15		
GO:0006935	chemotaxis	down 9	down 10	down 5	
GO:0007165	signal transduction	down 29			
GO:0005975	carbohydrate metabolic process	down 36			
GO:1901135	carbohydrate derivative metabolic process		down 35		
GO:1901564	organonitrogen compound metabolic process		down 127		
GO:0019538	protein metabolic process		down 69		
GO:0006412	translation		down 46		
GO:0006754	ATP biosynthetic process		down 8		
GO:0008610	lipid biosynthetic process		down 16		
GO:0046942	carboxylic acid transport			down 8	down 7
GO:0006355	regulation of transcription, DNA-templated				down 20

^a^ GO terms with padj < 0.05 were regarded as significant. ^b^ Number of up- or down-regulated genes associated with the GO term. Gene numbers in red, green and yellow indicate that the GO term contains, up-, down-, and up-and-down-regulated genes, respectively.

## Data Availability

All the available data are presented in the manuscript.

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
