# Peer review of "Disparate Effects of Two Clerodane Diterpenes of Giant Goldenrod (Solidago gigantea Ait.) on Bacillus spizizenii"

_ijms, 2024, doi:10.3390/ijms25031531_

Round 1

Reviewer 1 Report

Comments and Suggestions for Authors

Dear Authors,

The manuscript is well-written, and the inclusion of genome-wide transcriptomic analysis significantly strengthens the study. The findings have implications for understanding bacterial responses to diterpenes, and the paper is a valuable contribution to the field. However, I would still suggest some minor issues. While the introduction provides a good overview of the importance of novel antimicrobial compounds, it might benefit from a more explicit statement of the research gap and the specific objectives of the current study. Clearly articulating the research questions or hypotheses can enhance the introduction. There are a few instances where sentence structure and wording could be refined for clarity. For example, in the conclusions, some sentences are quite long, and breaking them down or rephrasing could improve readability.

regards, 

The other reviewer.

Author Response

Thank you for reviewer's recommendations.
We changed the appropriate parts of the manuscript according to the reviewer's recommendation.

1) While the introduction provides a good overview of the importance of novel antimicrobial compounds, it might benefit from a more explicit statement of the research gap and the specific objectives of the current study. Clearly articulating the research questions or hypotheses can enhance the introduction.

The introduction section was modified based on the reviewer's recommendation

2) There are a few instances where sentence structure and wording could be refined for clarity. For example, in the conclusions, some sentences are quite long, and breaking them down or rephrasing could improve readability.

We have modified the text of the conclusion section to improve readability.

Reviewer 2 Report

Comments and Suggestions for Authors

The practical application of the  investigated diterpenes (Sg3a and Sg6) should be mentioned in the conclusion part.

Figure 3 and figure 5 can be combined

Author Response

Thank you for reviewer's recommendations.

1) The practical application of the investigated diterpenes (Sg3a and Sg6) should be mentioned in the conclusion part.

We have supplemented the text with the following sentence about practical application of the investigated diterpenes: “The results support that the Sg3a and Sg6 diterpenes have outstanding antibacterial effect, thus they could be introduced as biopesticides or lead compounds after suitable chemical modification and formulation.”

2) Figure 3 and figure 5 can be combined

Since the two figures carry different information, one shows data on the number of genes with altered activity, while the other shows data on the common and specific genes of the different treatments, it is not advisable to combine them. Perhaps combining the two figures would make the presentation of the results more difficult, so we would refrain from this.

Reviewer 3 Report

Comments and Suggestions for Authors

The research is dealng with the activity of clerodane diterpenes isolated from Giant goldenrod root 2 (Solidago gigantea Ait.) on  Bacillus spiz-3 izenii.

There is a need to find new antibacterial substances to successfully treat pathogens in human, animal and plant, especially since there is increase in the resistance to the currently used compounds.

Here, the investigators have isolated new clerodane diterpenes with antibacterial activity against  range of Gram positive bacteria, and studied the effect of the  2 most active compound on gene expression of the bacteria, to understand their mode of action.

The MS is well written, very detailed, and has in-depth description of the results and the conclusions.  It is a continuation of their previous published work of isolation of the compounds.

I have few comments:

1. Abstract:

Please add a summarizing line, what the outcome of the results in general.

Line 29 : what do you mean “under other conditions”. Please add specification.

2.Introduction:

Line 65 and 77: In line 65-66 it is mentioned 8 previously isolated clerodane diterpenes by the same authors, where in line 77 it is mentioned 7 isolated clerodane diterpenes. Please clarify the difference. The same reference is mentioned for both lines.

Line 90-91- What do you mean by other condition? Other then what?

3. Results

Table 1: In the PDF file I have loaded, the line numbers are now within the first column of the table, making it difficult to follow what supposed to appear in this column  

I

Comments on the Quality of English Language

It seems well english written.

Author Response

Thank you for reviewer's recommendations.
We changed the appropriate parts of the manuscript according to the reviewer's recommendation.

  1. Abstract:

Please add a summarizing line, what the outcome of the results in general.

-We reorganized the abstract and added summary sentences at the end of the text.

Line 29 : what do you mean “under other conditions”. Please add specification.

Line 90-91- What do you mean by other condition? Other then what?

-In the text, the phrase "under other conditions" has been clarified to the phrase "under different environmental and nutritional conditions".

2.Introduction:

Line  65 and 77: In line 65-66 it is mentioned 8 previously isolated clerodane diterpenes by the same authors, where in line 77 it is mentioned 7 isolated clerodane diterpenes. Please clarify the difference. The same reference is mentioned for both lines.

-The discrepancy was caused by the fact that eight diterpenes were isolated in the original study, but one of them (Sg3c) was not available in sufficient quantities for subsequent experiments. We have corrected and supplemented text where necessary.

  1. Results

Table 1: In the PDF file I have loaded, the line numbers are now within the first column of the table, making it difficult to follow what supposed to appear in this column 

-Unfortunately, the row numbering has overwritten the first column in Table 1, but in the final version of the article, where the rows are not numbered, this will not cause a problem.